# Facilitating behavioral change: A comparative assessment of ASHA efficacy in rural Bihar

Oskar Burger[1]*, Faiz Hashmi[1], Maciej J. Dańko[2], Santosh Akhauri[3], Indrajit Chaudhuri[4], Emily Little[5], Hannah G. Lunkenheimer[1], Sudipta Mondal[4], Nachiket Mor[6], Neela Saldanha[7], Janine Schooley[8], Palash Singh[9], Tracy Johnson[10], Cristine H. Legare[1]

1 Center for Applied Cognitive Science, The University of Texas at Austin, Austin, Texas, United States of America, 2 Max Planck Institute for Demographic Research, Rostock, Germany, 3 Project Concern International, Patna, India, 4 Project Concern International, Delhi, India, 5 Nurturely, Bend, Oregon, United States of America, 6 Banyan Academy of Leadership in Mental Health at Kanchipuram, Chennai, Tamil Nadu, India, 7 Yale Research Initiative on Innovation and Scale, New Haven, Connecticut, United States of America, 8 Project Concern International, San Diego, California, United States of America, 9 SCOPE, Helsinki, Finland, 10 Bill & Melinda Gates Foundation, Seattle, Washington, United States of America

* oburger@utexas.edu

**Data Availability Statement:** All data used in this submission are publicly available without restriction via a dedicated Github page that can be

## Abstract

Community health worker (CHW) programs are essential for expanding health services to many areas of the world and improving uptake of recommended behaviors. One of these programs, called Accredited Social Health Activists (ASHA), was initiated by the government of India in 2005 and now has a workforce of about 1 million. ASHAs primarily focus on improving maternal and child health but also support other health initiatives. Evaluations of ASHA efficacy have found a range of results, from negative, to mixed, to positive. Clarity in forming a general impression of ASHA efficacy is hindered by the use of a wide range of evaluation criteria across studies, a lack of comparison to other sources of behavioral influence, and a focus on a small number of behaviors per study. We analyze survey data for 1,166 mothers from Bihar, India, to assess the influence of ASHAs and eight other health influencers on the uptake of 12 perinatal health behaviors. We find that ASHAs are highly effective at increasing the probability that women self-report having practiced biomedically-recommended behaviors. The ASHA's overall positive effect is larger than any of the nine health influencer categories in our study (covering public, private, and community sources), but their reach needs to be more widely extended to mothers who lack sufficient contact with ASHAs. We conclude that interactions between ASHAs and mothers positively impact the uptake of recommended perinatal health behaviors. ASHA training and program evaluation need to distinguish between individual-level and program-level factors in seeking ways to remove barriers that affect the reach of ASHA services.

## Introduction

Much of the world lacks sufficient healthcare and has high rates of infant and maternal mortality. One strategy for increasing access to care in service-limited regions is to introduce

found at this link: https://oskaratomni.github.io/ashaefficacydataarchive/.

**Funding:** The funding for this project comes from the Bill and Melinda Gates Foundation (BMGF): INV-008582. Supplementary funding was awarded via BMGF Grant: INV-016014. Both of these GMBF grants were awarded to CL (PI) at the University of Texas at Austin. TJ attended Project workshops and meetings as a BMGF Project Officer and team member with content expertise, which included contributing to discussions of methodology and project design. Aside from TJ's inputs in this role, the funders had no role in the study design, data collection, data analysis, decision to publish, or preparation of the manuscript. Administrative support for individuals based at UT Austin was provided by P2CHD042849, Population Research Center, awarded to the Population Research Center at The University of Texas at Austin by the Eunice Kennedy Shriver National Institute of Child Health and Human Development.

**Competing interests:** The authors declare that no competing interests exist for this manuscript.

community health worker (CHW) programs [1]. CHWs are typically selected from the communities they will serve and receive limited training. They have the goals of extending services but also of using their knowledge of local customs to help deliver more effective messaging, increase demand for key services, and facilitate behavior change [1]. Indeed, evaluations of CHW programs demonstrate that they are highly effective at promoting increased uptake of certain behaviors [2, 3]. However, some evaluations also find that some CHW programs are underperforming with respect to achieving certain levels of uptake in their beneficiary communities or argue for improvements in knowledge or particular skills [4].

In this paper, we focus on the largest CHW program in the world, the ASHA program (Accredited Social Health Activists), which was founded by the government of India's National Rural Health Mission (NRHM) in 2005 [5, 6]. The program started in the highest-need states of India in 2006 and gradually increased in scope, now serving nearly the entire country with over 1 million workers. In describing the ASHA role, the NRHM states that "every village/large habitat will have a female Accredited Social Health Activist (ASHA) . . . to act as the interface between the community and the public health system." As such, ASHAs often act as a bridge, as they are tasked with connecting local beneficiaries to the formal health system with the goal of improving maternal and newborn health outcomes [7]. At least in rationale, ASHAs are meant to facilitate behavior change, but this aspect of their role has arguably been de-emphasized and/or overshadowed in favor of service extension [8, 9].

The ASHA program has focused on behaviors especially relevant to reproductive, maternal, newborn, and child health (RMNCH), such as institutional delivery, the consumption of iron-folic acid (IFA) tablets, and antenatal and postnatal care visits. The goals of increasing uptake of such behaviors are to lower the high rates of maternal and infant mortality and morbidity that prevails in much of India, particularly in rural and hard-to-reach areas. For instance, Bihar, where the present study takes place, had the lowest per capita gross domestic state product in India in 2015, along with the lowest literacy rate and lowest real per capita revenue expenditure on health [10]. Bihar also has high levels of under-five mortality (58 per 1000 live births), infant mortality (42 per 1000 live births), and maternal mortality (208 per 100,000 live births) [10], all of which are higher than the averages across India and are of major concern from a global public health perspective. Motivated by indicators like these, considerable effort has gone into the development of public health initiatives and CHW programs in Bihar and across India.

As with other CHW programs, there are many indications that health outcomes have improved as a function of ASHA—mother interaction, especially if those interactions occur "early and often" [11]. For example, Smittenaar et al. [11] found several connections between ASHA presence or assistance and greater uptake of recommended health behaviors by recent mothers in Uttar Pradesh, India. A recent study by Agarwal et al. [5] found that ASHA interactions lead to increases in ANC checkups, institutional deliveries, and use of a skilled birth attendant. They also found that in areas where ASHAs were highly active, that women of the lowest castes and from the poorest households had higher odds of receiving ASHA services.

Other studies, however, have questioned the efficacy of ASHAs. Lyngdoh et al. [12] found that interactions with CHWs in India, including ASHAs, did not lead to increased knowledge about important aspects of perinatal health care. In a study focused on incentives, Koehn et al. [13] looked at four states in India and found that mothers were sometimes less likely to have adopted a recommended health behavior if they were visited by an ASHA rather than by an Anganwadi Worker (AWW; another type of CHW). Scott et al. [14] conducted an extensive literature review of 10 years of published research evaluating the ASHA program. In studies about routine ASHA performance, the majority of the reviewed papers found mixed or negative results (mixed: 55% (43 of 78); negative: 23% (18 of 78); less than 25% were positive). Indeed, when looking across the full range of studies of ASHA performance, it is easy to form

the somewhat contradictory impression that ASHAs are at once highly effective and not effective at all.

One shortcoming of ASHA evaluations is that many different metrics are used by different studies. For instance, across the studies summarized in Scott et al. [14], ASHA efficacy was defined differently based on the focus of each study, and included knowledge appraisals, opinions stated by mothers, self-reports by ASHAs, the frequency of ASHA contacts with mothers, and calculated effects of ASHAs on the uptake of one to several possible health behaviors, among other measures. Thus, across these studies, there was a wide range of definitions of efficacy. Because of such wide variation in how performance is evaluated, it is difficult to make general conclusions about ASHA performance or to identify which specific aspects of the ASHA role impact maternal health and behavior and why [11, 14]. Another issue with studies of ASHA efficacy is the lack of comparative reference. Many studies describe an ASHA's effect as positive, mixed, or negative, without considering the associated question of effective or ineffective compared to what/whom. For this reason, we consider ASHA efficacy compared to nine other possible health "influencers" who are known to convey health-relevant information in Bihar [15, 16].

Motivated by the inconsistent impression of ASHA performance due to the prevalence of mixed or negative evaluations and by the calls for more data on the matter, we analyze perinatal health data from 1166 recent mothers from Bihar, India. We define efficacy as the statistical effect of ASHAs on the uptake of biomedically-recommended health behaviors while acknowledging that efficacy can be captured with various possible metrics [14]. We look at efficacy across a range of nine potential sources of influence and 12 different perinatal-health behaviors and how they align with the biomedical recommendations that are intended to flow from the medical system to the community via CHWs. Finally, if service or performance gaps are found, it is important to consider potential barriers and offer a reasonable diagnosis of what is constraining the service. For this reason, we evaluate the possibility that differences between ASHAs and mothers due to caste, religion, or wealth affect the reach of services.

The data analyzed here are part of Project RISE, a mixed-methods research program aimed at understanding the many factors that influence how an ASHA affects maternal and newborn health in Bihar, India. Here we focus on the subset of our data that speaks to ASHA efficacy and health behavior.

### Research questions

Given recent interest on the topic of ASHA efficacy [5, 6], the general importance of CHWs as a potential agent of health-improving behavior [1, 2], and the impression from the literature that ASHAs have mixed or even negative effects on health behavior [14], we address the following questions:

Q1: What is the statistical effect of ASHAs on the uptake of recommended perinatal behaviors?

Q2: How does ASHA influence vary across behaviors, and how does it compare to other sources of influence?

Q3: Do caste, religion, or wealth act as barriers to effective service delivery?

To answer these research questions, we describe ASHA–mother contacts across a range of perinatal behaviors, quantify ASHA efficacy compared to other sources of influence and consider possible barriers to that influence.

## Materials and methods

### Participant sampling and recruitment

Project RISE investigators recruited and interviewed participants during a three-month period from June to August 2019. Bihar has three major language groups: Magahi, Maithili, and

Bhojpuri. The region where Maithili is spoken covers a much larger area of Bihar than the other two. For this reason, one district was sampled from the regions where Magahi and Bhojpuri are spoken, and two were sampled from the region where Maithili is spoken. Within each of these four districts, two blocks were randomly sampled, and from within each of the eight selected blocks, 50 Anganwadi Centers (AWCs) were randomly sampled. AWCs were the focal sampling unit because these represent the catchment areas for ASHAs. Three recent mothers were randomly recruited from each of these 400 AWCs (50 AWCs X 8 Blocks), based on ASHA registers. Thirty-four surveys were excluded for being incomplete or for falling outside of the recruitment criteria for the survey, resulting in a final sample size of 1,166. All of the mothers recruited had given birth within the previous six months to maximize memory of behaviors undertaken during the perinatal period.

Before interviewing the sample of mothers, we conducted focus group discussions (FGDs) with recent mothers and mothers-in-law in order to gain a wider qualitative understanding of perinatal behavior and belief. We also conducted one-on-one interviews with ASHAs and other local influencers (local priests, rural medical practitioners, other CHWs, and traditional birth attendants). These qualitative discussions provided valuable insights for the project and were also used to design the questionnaire. In the FGDs, we learned about many behaviors that are not usually the focus of health initiatives but were often referred to by the women who participated in the discussions. The FGDs informed the selection of behaviors in this study.

## Ethics statement

This project's methodologies, surveys, and consent procedures were reviewed and approved by the Institutional Review Board of the University of Texas at Austin (Study Number: 2018-01-0027; Approval Date: Feb/23/2018) and by Sigma Institutional Review Board in India (Study Number: 10056/IRB/D/18-19, Approval Date: Dec/22/2018). Participation in this study was strictly voluntary and informed consent was obtained for each respondent before the survey. The consent included a brief description of the survey, description of the role of the respondent in the study, including the expected duration of the respondent 's participation, clear indication that participation is voluntary, and that the information provided would be confidential. Consent was obtained verbally by each interviewer and recorded as a response on the form.

## Coding

**Focal behaviors.** Our extensive survey effort included questions about the uptake of perinatal behaviors, the type and nature of contact and services that ASHAs assisted mothers in obtaining, and about other factors related to perinatal decisions and behaviors. Some of these questions lead readily to a binary yes/no response, and others required coding. For example, one question asks about the month that ANC registration occurred. For the analysis below, this is considered a question about timely ANC registration and, as such, if the response is 3 months or less, it is coded as a 'yes' and if it is 4 months or longer, it is coded as a 'no.' As such, it is important to keep in mind that results in the 'not recommended' direction for ANC registration indicate late registration more often than not registering at all. Another question asks about treating the umbilical cord stump with a substance after delivery. The question has many options for locally-used substances (a locally-purchased 'blue medicine,' mustard seed oil, and talcum being the most common), but here we distill these responses to: 'treated the cord stump with something' (not recommended) or 'did not treat the cord stump' (recommended). Table 1 provides a definition for each of the 12 focal behaviors.

Three general types of behavior are included in Table 1. One group consists of perinatal health behaviors typically associated with the ASHA program that are directly incentivized:

**Table 1. Definitions and coding for each of the perinatal behaviors analyzed in this study.** The behaviors are ordered in approximate chronological order, with early pregnancy at the top of the table and postpartum behaviors at the bottom. The 'Recommended response' column indicates if doing the behavior was biomedically recommended or not.

| Behavior* | Recommended response | Definition |
|---|---|---|
| Conceal_Preg | No | Yes = concealed pregnancy three months or longer than three months; No = did not conceal or concealed less than two months |
| ANC_TimelyReg | Yes | Yes = registered within first 3 months of pregnancy |
| ANC_4checkups | Yes | Yes = had 4 or more checkups, no = had 0 to 3 |
| FastWhilePreg | No | Yes = did fast in some form, no = did not fast at all |
| WorkWhilePreg | No | Yes = frequently or sometimes, no = never |
| IFAtabs | Yes | Yes = consumed the full recommended amount; no = did not consume or consumed less than recommended |
| Hospital_del | Yes | Yes = government or private hospital; no = home birth |
| FeedColostrum | Yes | Yes = fed colostrum; no = did not feed colostrum. |
| TIBF_onehr | Yes | Yes = breast fed within first hour; no = after first hour or never breastfed |
| Cordstump_Apply | No | Yes = applied something to cord stump (but there are many options for what was applied); no = applied nothing. |
| Bath_24hpp | No | Yes = gave bath within 24 hours; No = within 24 to 48 hrs, after 48, and bath not given. |
| AvoidCereal | No | Yes = avoided cereals just after delivery |

*Abbreviated names, as used to label Figure axes, below, where space does not permit writing out a full description of the behavior.

ANC registration, ANC checkups, and institutional delivery. A second group includes behaviors that are indirectly incentivized: consuming 100 or more IFA tablets during pregnancy, colostrum feeding, timely initiation of breastfeeding (TIBF), care of the cord-stump, and not bathing the newborn within 24 hours. The difference between directly and indirectly incentivized is whether or not the ASHA receives a payment specifically for completing that discrete task (e.g., an ASHA *should* receive a payment for each documented institutional delivery).

Indirectly incentivized tasks are linked to messaging and counseling duties that the ASHA has and therefore do not lead to payment on a per-completion basis. For instance, the ASHA is meant to relay information that encourages taking IFA tablets and feeding colostrum, but she does not receive a separate payment for each woman who feeds colostrum to her newborn, nor does she complete paperwork registering that a woman in her catchment self-reports feeding her newborn the colostrum. IFA tablets are indirectly incentivized based on the ASHA distributing the target amount but not on what individual mothers report actually consuming them.

The third group of behaviors evaluated are not incentivized nor typically associated with the ASHA program. These behaviors were identified during our qualitative interviews and included concealing the pregnancy, fasting during pregnancy, doing heavy work during pregnancy, and avoiding cereal-based foods in the first week postpartum. These were mentioned during focus group discussions with mothers and mothers-in-law and then added to our quantitative survey for exploratory purposes. We found that many women spontaneously associated advice on these behaviors with local influencers, including ASHAs. Concealing pregnancy, particularly early pregnancy, is an extremely common behavior, practiced in most cultures of the world. Here it is coded as 'not recommended' because it can result in delays in ANC registration and the initiation of other perinatal health behaviors [17], such as the start of taking IFA tablets. In this sense, we are not implying that pregnant women should disclose their new pregnancies to the general public by any given time, but rather that a default biomedical recommendation in this setting would be that mothers disclose a pregnancy to relevant health officials, especially the ASHA, as early in the pregnancy as possible because this facilitates earlier ANC registration and may increase the probability of taking the full recommended dosage of IFA. We also found that many women reported fasting during their pregnancy, either

regularly or for festivals. From a biomedical perspective, food limitation during pregnancy is not recommended, especially as women in Bihar are commonly under-nourished at the start of pregnancy [18, 19]. We also included mothers avoiding cereal-based foods postpartum as a behavior that is not biomedically recommended, with the same reasoning that the caloric demands of breastfeeding call for an increase in food intake and, as such, food restrictions in a food-limited context should be avoided when the body is producing breastmilk. Moreover, cereals are the most dominant food group consumed by women in Bihar, where dietary diversity tends to be low [19], and where there are many existing social conventions that delay or prevent women from eating [20]. The avoidance of cereals is related to a commonly practiced postnatal ritual called Chhathi, which occurs on the sixth day after birth. Some versions of Chhathi call for avoiding cereals between delivery and the occasion on the sixth day after birth. This third group of behaviors provides interesting reference points for the range of behaviors that mothers in this sample reported as relevant to their perinatal experiences and that they associated with various local health-relevant influencers.

Each of the behaviors in Table 1 is analyzed with respect to biomedical recommendations. For some behaviors, the recommendation is to do the behavior and, in these cases, 'yes' is the recommended response. For others, the recommendation is to not do the behavior, and 'no' is the recommended response. For the analysis below, these latter cases where 'no' is the recommended response are reverse coded such that the question is 'did the participant do the recommended behavior?'

**Influencers.** While the ASHA is the focus of our analysis in this study, it is important to note that ASHAs are one of many possible influencers who can lead to health-relevant decision-making. Within India's CHW workforce, there are two main categories of CHWs, namely ASHAs and Auxiliary Nurse Midwives (ANMs), with extended support of AWWs through the Integrated Child Development Services system. Across these CHWs, ASHAs, ANMs, and AWWs each have distinct but complementary responsibilities. ASHAs were meant to be health promoters, activists, and counselors (but the degree to which the program has continued to emphasize all of these roles is debatable). ASHAs are also the first points of contact for linking the beneficiaries to appropriate health services. While the AWW and ASHA have distinct roles, there are also many overlaps related to maternal and child health and nutrition. In addition to these other CHWs, family members, friends, traditional birth attendants (Dais), and others can influence maternal decision making [21]. Most of these sources, and their interrelationships, have not been widely studied nor compared specifically to ASHA influence.

For these reasons, each of the behaviors in Table 1 can be influenced by nine possible sources (options that were selected by only a few women were collapsed into a category of "OTHER" influence). These sources of influence for each behavior are described in Table 2.

During the survey, each participant is asked about the sources of influence regardless of the response. That is, we ask participants who the sources of influence were for those who had a hospital delivery as well as those who did not. For each of these behaviors, participants spontaneously mentioned the sources of influence, which were recorded to a list of likely options that had been pre-populated (based on pre-survey focus group discussions and piloting) using data collection tablets. This aspect of the survey design is important to keep in mind in light of the results below because participants were *not* directly asked something akin to *Did the ASHA influence your decision about this behavior*? Rather, they were asked *who influenced you to do this*? To which respondents named sources of influence spontaneously and without prompting. Hence, mentions of the ASHA in association with behaviors that are incentivized and very clearly part of the program, as well as those that are not typically associated with the ASHA, can reveal the salience of the ASHA's association with a range of behaviors and the observed mentions are not a function of being primed by the survey question.

**Table 2. Descriptions for each influencer category assessed in the analysis below.**

| Influencer* | Description |
|---|---|
| ASHA | Accredited Social Health Activist, a government-trained female CHW selected from the community. ASHAs work as an interface between the community and the public health system. |
| ANM or AWW | Auxiliary Nurse Midwife (ANM), a type of CHW based at a health sub-center or Primary health center. ANM's responsibilities include family planning, immunization, infectious disease prevention, and care, in addition to maternal health and childbirth. Anganwadi worker (AWW), is a CHW based at the village level who is primarily tasked with distributing food and nutrition supplements. |
| GovDoc | Government Doctor, a registered medical practitioner who works at public hospitals. Government doctors are responsible for the proper functioning of the hospital and activities in relation to NRHM and other National Programs. |
| Privclinic | Private clinic, a clinic associated with the non-governmental (private) medical system. |
| RMP | Rural medical practitioner, an individual often associated with local pharmacies. In many parts of Bihar these are the nearest option for health advice or basic services. They are non-certified and of variable quality. |
| Fam | Family, a member of the family living in the mother's household, primarily husband and mother-in-law. |
| no_one | No one, a category on the survey indicating that the respondent did not readily associate the behavior, nor their proclivity to do it, with any specific source of influence. |
| Friendrelneigh | Friends, relatives, or neighbors, a source of influence in the immediate vicinity of the respondent but outside the household. |
| Dai | Dai, a traditional birth attendant who is common and culturally important in Bihar and many other areas of India. |

*Abbreviated names, as used to label Figure axes, below, where space does not permit writing out a full description of the behavior.

## Analysis procedure

For question one (Q1: What is the effect of ASHAs on the uptake of recommended perinatal behaviors?), we measure the ASHA effect on maternal behaviors with two approaches. For the first, the response variable is a maternal health score that captures the uptake of recommended behaviors by mothers. We examine the relationship between this health score and an ASHA interaction score that captures ASHA-mother interaction as a sum of all visits and services provided that were captured in our dataset. The **maternal health score** is a simple count of the number of health behaviors each mother reports adopting of the 12 in Table 1. In some cases, the recommendation is to do a behavior (e.g., have hospital delivery, register for ANC, feed colostrum), and in other cases, the recommendation is to not do a behavior (e.g., do not treat the cord stump with a substance, do not bathe the newborn with 24 hours of delivery, do not engage in heavy labor during pregnancy, do not fast during pregnancy). For this reason, we code all questions as a 1 if the response is consistent with the biomedical recommendation and a 0 if it is inconsistent.

The **ASHA interaction score** is a sum of all specific named interactions and services associated with ASHAs. This includes the number of household visits made by trimester, the number of postpartum visits, and the number of services associated with ASHAs (such as if an ASHA was named for assisting with immunization, filaria, receiving a payment, obtaining the sufficient supply of IFA tablets). A list of the items from the questionnaire that contributes to this score and the number responding in the affirmative is in SOM (S1 Table).

We investigated the correlation between these two measures with a regression model where the maternal health score was the response variable. This was first modeled as a Poisson distribution because the response is a count variable, but significant under-dispersion was detected.

We then compared fits of a negative binomial, a quasi-Poisson, and a Mean Parametrized Conway-Maxwell Poisson (MPCMP, [22]) by AICc. The MPCMP had the lowest AICc value and was used in the results below. The coefficients were identical across all four models (Poisson, Quasi-Poisson, Negative Binomial, and MPCMP); the choice of error distribution affected only the standard error (and hence P-value).

The second analysis used to address Q1 uses a measure of ASHA effect based on the follow-up questions about influence (e.g., across all behaviors, the survey asks 'who influenced your decision to do (or not do) this?'). The response variable is binary, indicating a yes/no for whether or not each response was aligned with the biomedical recommendation for each behavior. We used logistic regressions to estimate odds ratios that women report adopting a recommended behavior given the source of influence associated with it. The control variables in this regression are **age** (categorical), **age at marriage** (categorical), **parity** (categorical and based on the number of total children at the time of interview), **education** (categorical), **wealth** (categorical, measured as quintiles based on a principle component analysis of a multi-item questionnaire). To compute an individual wealth score, we used a 36-item inventory that includes questions about possessions and household characteristics (the wealth measure is well-vetted and used by the National Family Health Survey, [23]). The resulting wealth measure are based on a principle component analysis (Base R function prcomp, R Core Team (2021)).

For Question 2 (Q2: How does ASHA influence vary across behaviors and how does it compare to other sources of influence?), we again use a binary response variable for the uptake of recommended biomedical practices, but we also include interactions between each influencer and behavior. We then use a model selection algorithm to remove irrelevant interactions.

The model selection process worked as follows: We start with a saturated model that includes all possible behavior by influencer interactions and then use a backward model selection procedure [24] to sequentially drop interactions to optimize model performance measured by AICc (Second-order Aikike Information Criteria) [25]. Interactions that do not minimize AICc are dropped from the model. This sequential dropping of irrelevant interactions eventually results in the most parsimonious model. Using AICc as a criterion for excluding terms helps discourage over-fitting, which is desirable because increasing the number of parameters in the model almost always improves the goodness of the fit. The final selected model is then used to estimate predicted probabilities that mothers engage in each behavior. By using predicted probabilities computed from the final model, we include all retained interactions as well as controls in the estimated probabilities. Controls were included in addition to the categorical variable for each question and a dummy variable for each influencer. Participants could report multiple influencers for each question.

To validate the final selected model, we calculated variance inflation factors (vif) using the R package car [26]. If individual variables have large effects inflating the variance of the model, then they may be problematically correlated with other variables in the model. We examined the final model for vifs and sequentially removed interactions with the highest vif until all were below 5.0 (which resulted in the removal of one interaction, between ASHA influencer and the behavior of early ANC registration.) All analysis was conducted with the R open-source computing software [27].

For Question 3 (Q3: Do caste, religion, or wealth act as barriers to effective service delivery?), we conduct a moderation analysis to evaluate if caste, religion, or wealth affect ASHA service delivery, using a separate model for each possible moderator. Moderation analysis tests for the possibility that a relationship between two variables is modified by a third. We ask if caste, religion, or wealth moderate the relationship between maternal health score and ASHA interaction. To do this, we added interaction terms to each potential moderator resulting in

three separate models, one for each possible moderator. Differences in religion and caste were coded as binary variables where 1 indicates ASHA and mother are from the same caste or religion and 0 means different caste or religion. Wealth difference is a continuous measure based on subtracting the wealth score of the mother from the wealth score of the ASHA. In fitting these models, significant under-dispersion was also detected, and we again fit MPCMP generalized linear models using the R package mpcmp [22]. The test for moderation is based on a significant interaction between each moderator and the ASHA interaction score. Both the interaction and the main effect must be considered to understand the result of the moderation. We use a graphical approach to do this.

## Results

### Composite scores for ASHA interaction and maternal health

The analysis for Q1 and Q3 considers two scores that are composites of variables recorded in our survey of recent mothers: the ASHA Interaction Score and the Maternal Health Score. Both are count variables. The ASHA Interaction Score is a tally of all ASHA interactions and service-associated references, and the Maternal Health Score is a count of each of the behaviors in Table 1 that is done according to biomedical recommendations.

Summing the responses that compose the ASHA interaction score yielded an integer value that varies from zero to 74 with a median of 9. The largest three values, 74 and two at 60, are extreme values reported by women who ASHAs visited 20 and 30 times in the second and third trimesters. These unusually frequent visits occurred because these women lived next door to an ASHA. They were excluded from the analysis below.

The Maternal Health Score is a count of each behavior that the mothers report having practiced according to biomedical recommendations during her last pregnancy. The maximum possible value for this measure is 12, one for each behavior in Table 1. The mean was 6.35, and the median was 7.0. Density plots for each measure are in Fig 1.

### Independent variables and descriptives

Descriptive statistics for the control variables are listed in Table 3. All variables are categorical, so we report the counts and percentages of samples in each category. Across the control

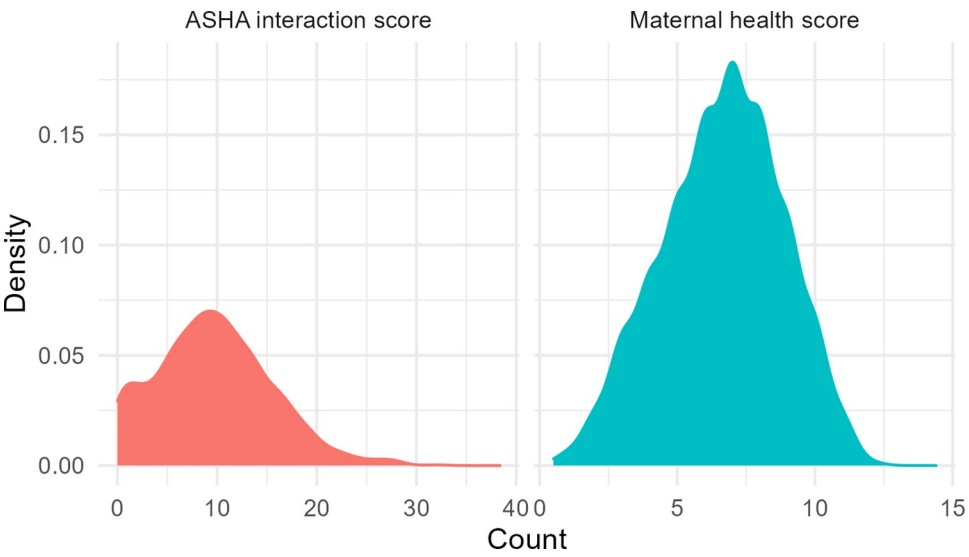

**Fig 1.** Density plots for ASHA Interaction Score (left) and Maternal Health Score (right).

**Table 3. Descriptive statistics for categorical control variables used in the regression models below.**

| Characteristic | N = 1,166[1] |
|---|---|
| **Age** | |
| 18–20 | 296 (25%) |
| 21–24 | 421 (36%) |
| 25–28 | 297 (25%) |
| 29–33 | 105 (9.0%) |
| 34+ | 47 (4.0%) |
| **Age at Marriage** | |
| 10–14 | 156 (13%) |
| 15–17 | 493 (42%) |
| 18–20 | 454 (39%) |
| 21+ | 63 (5.4%) |
| **Parity** | |
| 1 | 310 (27%) |
| 2 | 316 (27%) |
| 3 | 284 (24%) |
| 4 | 158 (14%) |
| 5+ | 98 (8.4%) |
| **EDU** | |
| 0 | 572 (49%) |
| 1–7 | 142 (12%) |
| 8–10 | 280 (24%) |
| 11–13 | 100 (8.6%) |
| 14–17 | 72 (6.2%) |

[1]n (%)

variables, the most common values in each are: aged 21 to 24, married between ages 15 and 17, two children, and no education.

We also describe the nature and frequency of ASHA–mother interactions to provide a more fine-grained breakdown of the ASHA Interaction Score. For instance, the number of home visits during pregnancy (ANC visits) was highly variable. Many mothers did not receive a single home visit for the duration of a trimester. Of the women surveyed, 253 (21.6%) did not receive an ASHA visit during pregnancy. The number of women going a full trimester without an ASHA visit decreases from the first to the third trimesters (402 did not receive a visit in the first trimester, 207 did not in the second trimester, and 160 did not in the third trimester).

Another question on the survey asked about postnatal care (PNC) visits during the first week after birth. Of the women surveyed, 615 (52.5%) did not receive a postpartum home visit in the first week after birth.

The survey also asked mothers about visits with the ASHA outside of the home or if the ASHA sends her messages through a family member. We found that 46% of recent mothers said they had a health-related visit with an ASHA outside of their home, and 39% said they received health-related advice that an ASHA delivered to them via a close relative. Moreover, 23% said yes to both of these questions. These alternative routes may be an important method for ASHA messaging (and influence).

Of the mothers who did not receive a home visit during pregnancy, 77 met with the ASHA outside of their homes, leaving 176 (15%) with no direct ASHA contact during pregnancy.

Lastly, we also ask mothers if they received health-related messages from their ASHA that was sent through a family member, of which 457 (39%) did. However, there were still 154 (13.2%) recent mothers who did not receive a home visit, did not meet with an ASHA outside of their homes, nor receive health messaging from the ASAH via a family member.

Some evaluations of maternal health behavior look at 'bundles' of actions that fit together. For instance, Kumar et al. [28] define the 'Full ANC' as 100 or more IFA doses, at least one tetanus injection (TT), and four or more ANC checkups. While tetanus injection is not a focal behavior of this study, we asked mothers about receiving an injection in our survey and can compare the percent of mothers in our sample who got the 'full ANC' to the results of Kumar et al. (2019). In short, our results are very similar. We find that about 23% of mothers reported adopting all three of these behaviors, while they found 21% (using data from India's National Family Health Survey 4). Moreover, the percentages adopting each component of the full ANC (IFA regimen, one TT injection, and 4 or more ANC visits) are very similar in our sample from Bihar and the national averages in Kumar et al. [28]).

## Q1: What is the statistical effect of ASHAs on the uptake of recommended perinatal behaviors?

As described in the Methods, we fit two statistical models to assess Q1. The first uses a measure of ASHA interaction, and we fit a MPCMP model to assess the relationship between maternal health score and ASHA interaction score (Fig 2). Increased interaction with ASHAs was associated with the uptake of more recommended perinatal health behaviors such that each

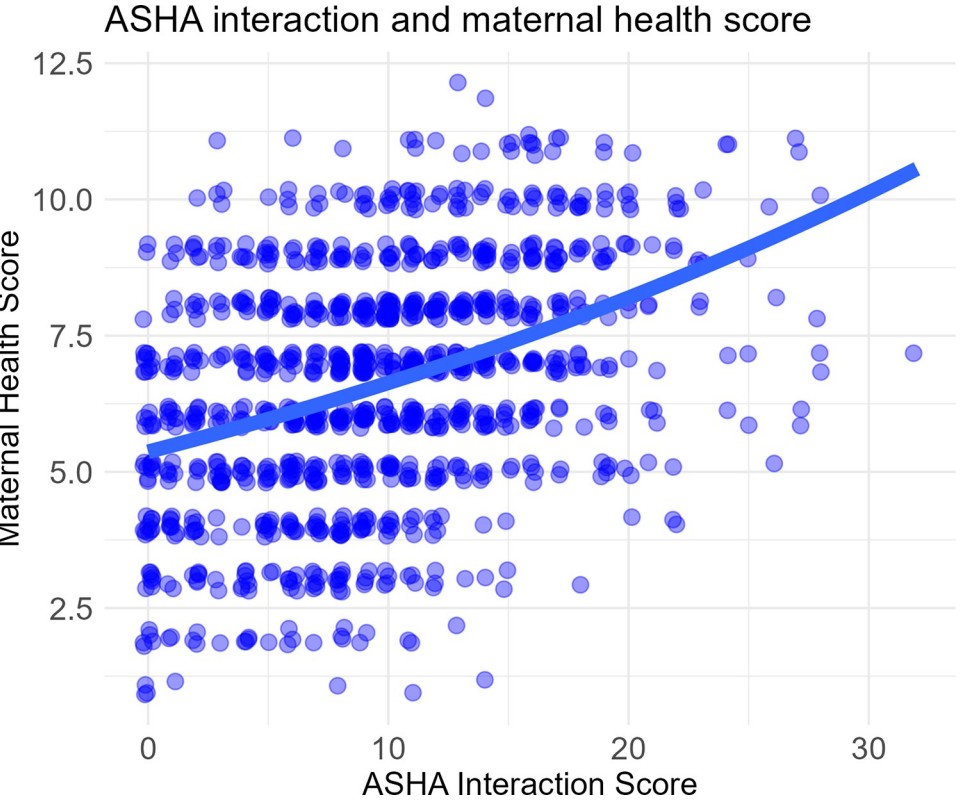

**Fig 2. Maternal health score as a function of ASHA interaction score.** All data points are integers. Their positions have been jittered for visibility.

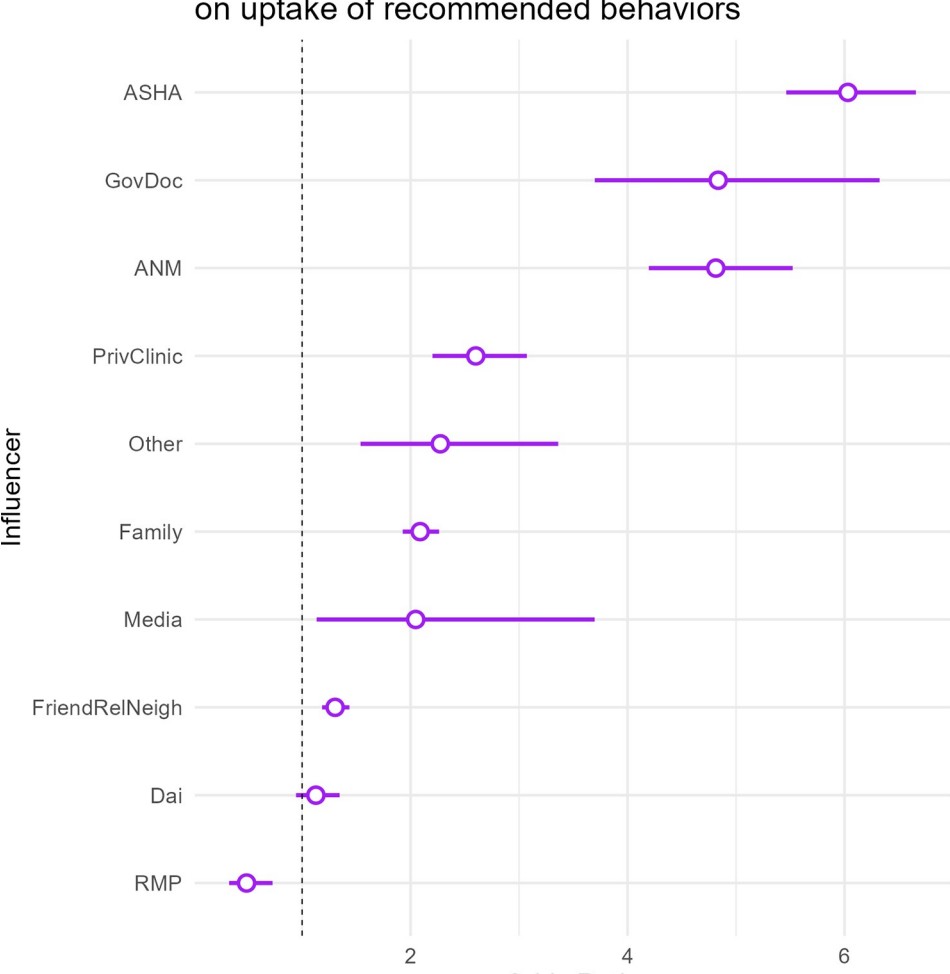

**Fig 3. Odds ratios for each influencer on overall uptake of biomedical behaviors.** See SOM for the associated table of results for this regression (S2 Table).

additional interaction increases the maternal health score by about 2% (IRR = 1.021, 95% CIs: 1.018–1.025). Across the range of ASHA Interaction Score, there is wide variation in uptake, and many women have very low scores.

The second model used to assess Q1 consists of a binary response variable with controls for age, wealth, education, and parity, along with dummy variables for each source of influence (see Analysis Procedure, above). Fig 3 shows that ASHAs have the largest positive effect on the overall uptake of health behaviors among the influencers considered here (OR = 6.03, 95% CI: 5.46–6.66, the reference category is no one). The only influencer with an overall effect on the uptake of recommended behaviors that is lower than the reference category of 'no one' is the RMP (OR = 0.49, 0.32–0.72), meaning that if an RMP was named as an influencer, mothers were roughly 50% less likely to have taken up biomedically-recommended behaviors than if they named 'no one' (note that naming no one was over six times less likely than those who named ASHA). All other influencers are positive or neutral but note that the selection of 'no one' as a source of influence seems to have been particularly common among women who did

not do the behavior in the recommended direction (a table of coefficients for the model presented in Fig 3 is in the SOM, S3 Table).

## Q2. How does ASHA influence vary across behaviors and how does it compare to other sources of influence?

To infer which influencers had the strongest effects on changing the probability that mothers engaged with each recommended behavior, we report model-predicted probabilities that mothers adopt each recommended behavior given each source of influence, accounting for controls and the most parsimonious set of interactions between influencers and behaviors (Fig 4). The values in each cell of Fig 4 are predicted probabilities based on the final selected model (the most parsimonious model as determined by AICc, see methods). As such, each cell represents the probability that a mother engages in each behavior with all controls at reference values (A figure reporting percentages for each influencer associated with reach behavior can be found in the SOM, S1 Fig).

### Model-predicted probabilities

| Behaviors | GovDoc | ASHA | ANM/AWW | PrivClinic | Media | Other | RMP | Family | FriendRelNeigh | No one | Dai |
|---|---|---|---|---|---|---|---|---|---|---|---|
| Conceal_Preg | 0.65 | 0.23 | 0.05 | 0.71 | 0.34 | 0.27 | 0.23 | 0.1 | 0.14 | 0.13 | 0.08 |
| ANC_TimelyReg | 0.67 | 0.74 | 0.25 | 0.25 | 0.36 | 0.29 | 0.25 | 0.39 | 0.29 | 0.14 | 0.08 |
| ANC_4checkups | 0.73 | 0.54 | 0.58 | 0.85 | 0.43 | 0.35 | 0.31 | 0.46 | 0.42 | 0.17 | 0.11 |
| FastWhilePreg | 0.92 | 0.88 | 0.86 | 0.94 | 0.77 | 0.27 | 0.66 | 0.41 | 0.27 | 0.48 | 0.35 |
| WorkWhilePreg | 0.87 | 0.72 | 0.43 | 0.9 | 0.66 | 0.58 | 0.53 | 0.69 | 0.41 | 0.35 | 0.24 |
| IFAtabs_full | 0.37 | 0.43 | 0.33 | 0.54 | 0.02 | 0.1 | 0.09 | 0.15 | 0.14 | 0.04 | 0.03 |
| Hospital_del | 0.87 | 0.9 | 0.78 | 0.9 | 0.66 | 0.58 | 0.53 | 0.68 | 0.65 | 0.35 | 0.24 |
| FeedColostrum | 0.86 | 0.94 | 0.92 | 0.71 | 0.86 | 0.82 | 0.79 | 0.59 | 0.66 | 0.64 | 0.71 |
| TIBF_onehr | 0.69 | 0.83 | 0.84 | 0.13 | 0.38 | 0.3 | 0.27 | 0.41 | 0.37 | 0.15 | 0.57 |
| Cordstump_Apply | 0.59 | 0.7 | 0.76 | 0.2 | 0.76 | 0.69 | 0.06 | 0.21 | 0.23 | 0.46 | 0.2 |
| Bath_24h_postpart | 0.97 | 0.88 | 0.93 | 0.97 | 0.89 | 0.85 | 0.82 | 0.56 | 0.54 | 0.68 | 0.56 |
| AvoidCereal | 0.79 | 0.92 | 0.86 | 0.28 | 0.93 | 0.91 | 0.89 | 0.72 | 0.78 | 0.79 | 0.4 |

fit
0.75
0.50
0.25

**Influencers**

**Fig 4. Model-predicted probabilities of doing the recommended behavior for each combination of influencer and behavior (given controls and the most parsimonious set of interactions among influencers and behaviors).** Questions, where the recommendation is to not do a behavior, have been recoded such that a response of 1 always means consistent with biomedical recommendations (e.g., 1 = did not work during pregnancy, 1 = did not give a newborn a bath within 24 hours, 1 = did feed baby colostrum, etc.) Behaviors are listed roughly sequentially from top to bottom on the y-axis. Influencers are ordered by the average magnitude of effect from left to right on the y-axis such that more strongly positive sources of influence are to the left. See SOM for full results of the underlying regression (S3 Table).

Behavior by behavior, the ASHA is consistently among the influencers with the highest positive effect on the probability that recent mothers are engaging in recommended health behaviors (Fig 4). For instance, ASHAs have the strongest positive effect on the probability that mothers had an institutional delivery (0.9 predicted probability if ASHA was named as an influencer) and early ANC registration (0.74 predicted probability of early ANC registration if ASHA named as an influencer). For fasting while pregnant and working while pregnant the ASHA was second and third respectively.

Many women did not receive a home visit in the first week postpartum, but the ASHA has a strong positive effect on behaviors during that time (feeding colostrum, TIBF, treating the cord stump). The ASHAs effect is relatively strong and positive for directly and indirectly incentivized behaviors as well as those that are not incentivized. The predicted probability that a mother completes a full IFA regimen is seemingly low if she names an ASHA as an influencer, at just .43, but that is the second-highest value across influencers for that behavior and IFA completion may be limited by supply issues that are out of the ASHA's control.

Fig 4 gives the model-predicted probabilities of doing the recommended behavior for each influencer category (given control variables and the retained interactions). We can also use the selected model to calculate inferential statistics for which influencers are raising or lowering the probabilities. We estimate the effect of each influencer by behavior during pregnancy (Fig 5) and postpartum (Fig 6). The panels in these two Figures identify the magnitude each influencer has the probability of doing the recommended behavior relative to all other sources of influence. The thin black horizontal dotted line indicates the reference level of no one. The numbers in italics for each category on the x-axis give the number of times that influencer was named for that behavior. For example, the upper right panel of Fig 6 shows ANC registration, and 521 women named the ASHA as an influencer for this behavior, 366 named family, 200 named friends/relatives/neighbors. This is useful for identifying cases where rarely mentioned influencers are having large effects on the predicted probability or when commonly named influencers have intermediate effects. For instance, the media have a large positive statistical effect for not applying substances to the cord stump, as recommended, but the media was mentioned just six times in association with this behavior.

Across the behaviors considered here, the ASHA is consistently nudging behavior in the biomedically-desirable direction, and her effect is always more positive than sources of normative influence like family or friends/relatives/neighbors (Figs 5 and 6). Her effect is typically near the formal health care providers (government doctors and private clinics), and sometimes even larger in magnitude (e.g., hospital birth, feeding colostrum). We see clear evidence that she is having positive effects on behaviors of biomedical relevance that are not formally associated with the ASHA role (concealing the pregnancy, working while pregnant, fasting while pregnant, and avoiding cereal-based foods in the first week postpartum).

## Q3. Do caste, religion, or wealth act as barriers to effective service delivery?

To address Q3 we conducted a moderation analysis to see if the influence of ASHA interaction on the number of health behaviors adopted by mothers was moderated by ASHA-mother differences. We considered differences by caste (binary), religion (binary), and wealth (continuous, and mean-centered). None of these differences removed the strong positive effect that ASHA interaction has on the number of health behaviors adopted, but the interaction between caste and ASHA interaction score was significant and the interaction with religion was nearly so (regression tables and plot of odds ratios in SOM; S2 Fig, S4 Table). However, the main effect of caste difference was slightly positive and the interaction was slightly negative, yielding a total effect of caste difference not being a major impediment to ASHA-mother contacts.

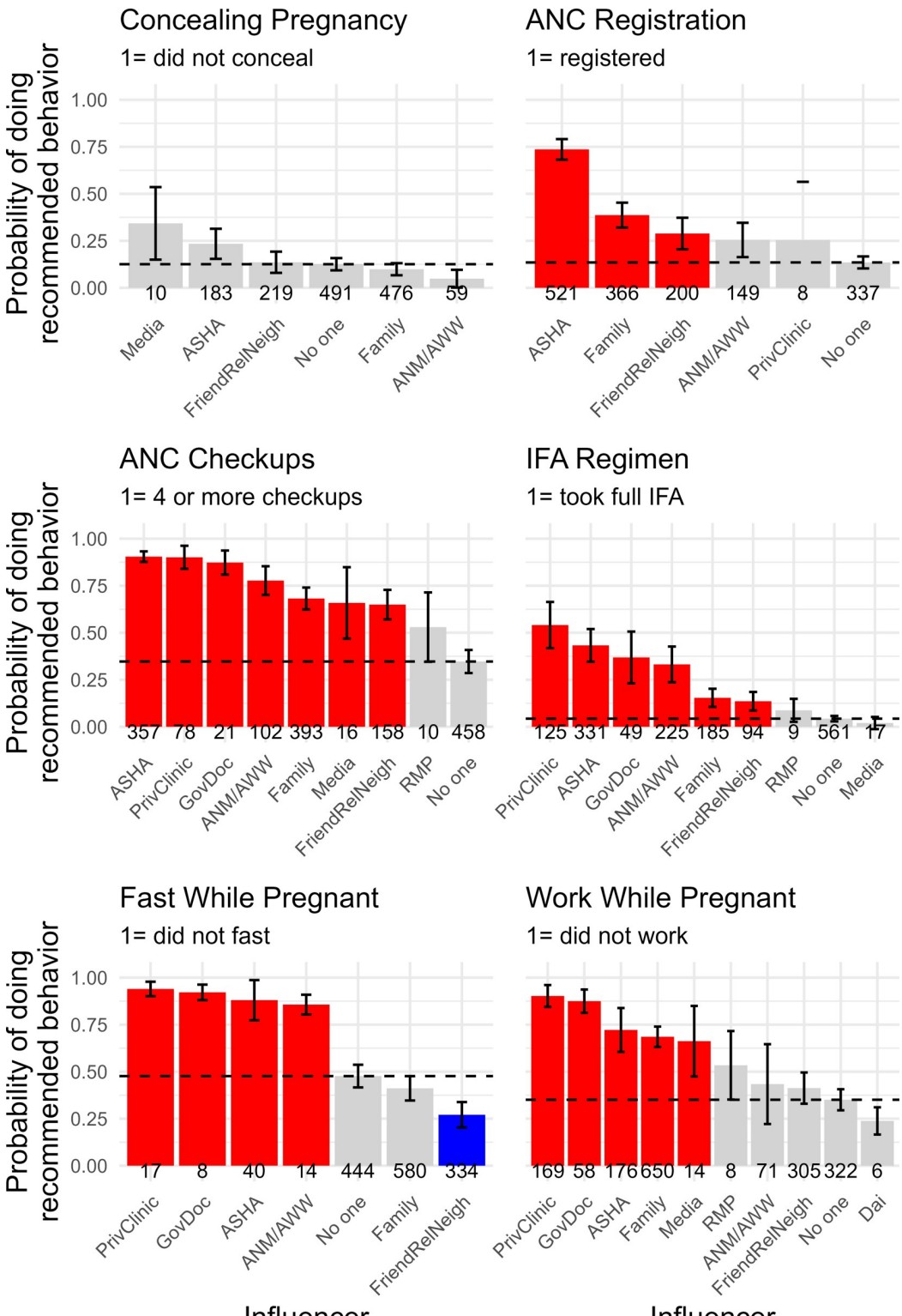

**Fig 5. Predicted probabilities for each behavior during pregnancy.** Red bars are positive effects (confidence intervals that are above and do not overlap with the reference category of no one). Grey bars are effects similar to no one (confidence intervals overlap). Blue bars are negative effects, meaning the influencer lowers the probability that the woman practices the recommended behavior compared to the reference category of no one. The thin black dashed line is drawn at the predicted probability of the reference category, 'no one'.

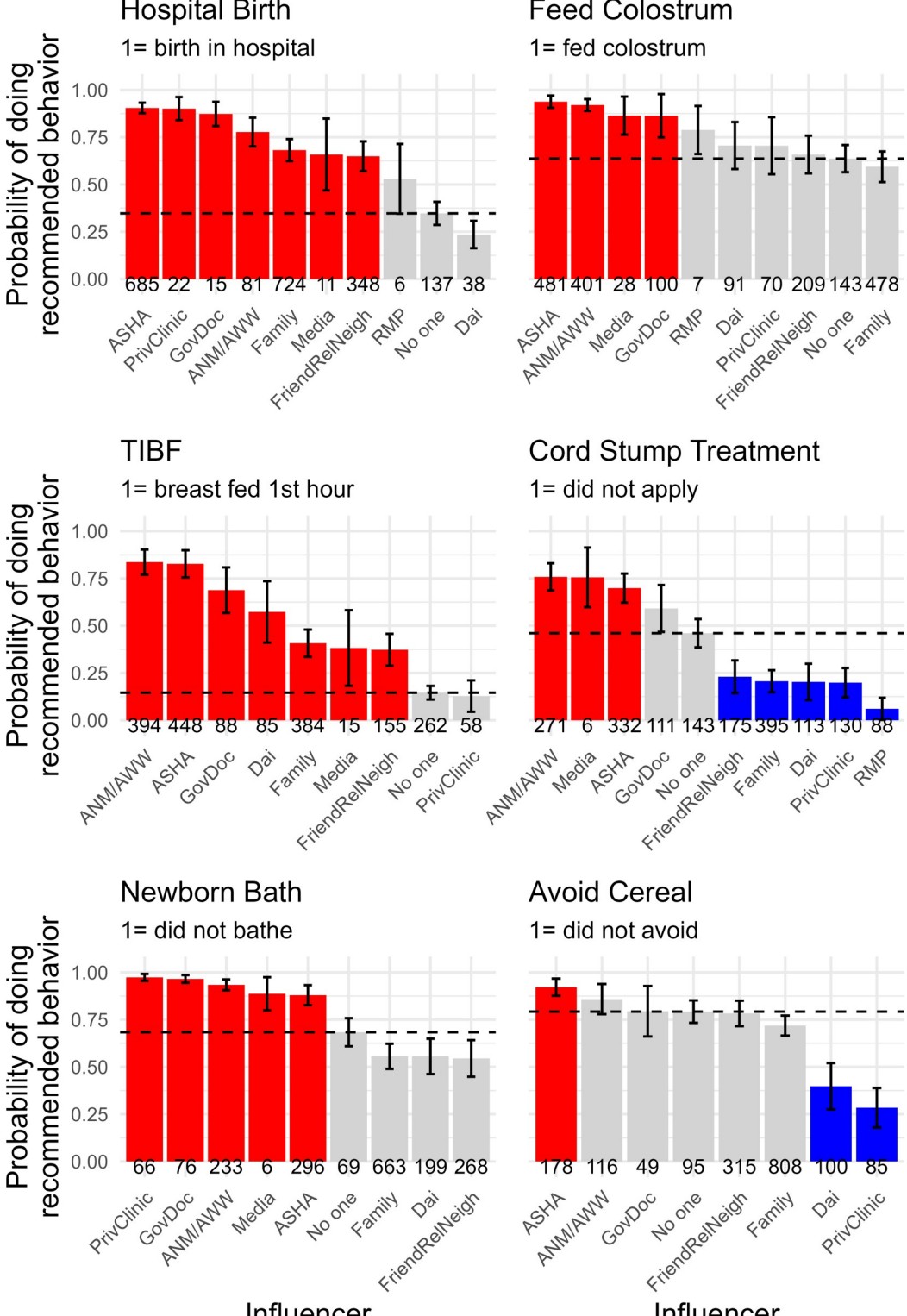

**Fig 6. Model predicted probabilities of doing each recommended behavior postpartum.** Red bars are positive effects (confidence intervals that are above and do not overlap with the reference category of no one). Grey bars are effects similar to no one (confidence intervals overlap). Blue bars are negative effects, meaning the influencer lowers the probability that the woman practices the recommended behavior compared to the reference category of no one. The thin black dashed line is drawn at the predicted probability of the reference category, 'no one'.

To better visualize the moderating effects of religion and caste, we plot the predicted effects of each (Fig 7). This shows that the effects of each are indeed very minor, and do not likely account for the barrier that prevents some mothers from receiving home visits during pregnancy. For caste, the effect is such that when an ASHA and mother castes are different, the maternal health score raises more steeply with ASHA interaction than when they are the same (a finding similar to [5]). For religious differences, there is not an appreciable effect.

We conducted some exploratory supplementary analysis to further evaluate if differences between ASHAs and Mothers in caste, religion, or wealth, could be factors that limit ASHA efficacy or reach. In these, we identified an 'at risk' group of women as those who did not receive any home visits during pregnancy from the ASHA (described above under "Independent Variables and Descriptives"). We used the same set of controls and fit a binomial logistic regression in an attempt to identify factors that might predict membership in this 'at-risk group' (the response variable was simply 1/0 for did not receive a home visit/did receive at least one home visit). The results suggest that caste differences actually reduce the risk of not receiving a visit (OR = 0.706, 95% CI: 0.529–0.942) and wealth differences may slightly increase the risk of being in this group although the CIs do encompass 1.0 (OR = 1.117, 95% CI: 0.989–1.264). However, these results are sensitive to how the risk group is defined and do not paint a clear picture for the role of these differences in affecting those women who receive the fewest ASHA contacts. For instance, the effects become neutral if we define the risk group as women who did not receive a home visit and did not receive a visit from the ASHA outside of the home. The details of these analyses are in the SOM (S5 Table). In this analysis the signal of such differences was either very mild or not detectable and other factors should also be considered when evaluating the observation that some mothers did not receive home visits.

## Discussion

### ASHAs are highly effective at increasing uptake in 12 health-promoting behaviors

Our results indicate strong positive correlations between the uptake of recommended behaviors by mothers and either having contact with ASHAs or being influenced by them. Mothers who named ASHAs as a source of influence were more than six times more likely to adopt a recommended behavior than those who named 'no one' (Fig 3). When individual behaviors are examined (Figs 5 and 6), ASHAs have a strong positive association with all 12 focal behaviors (with the exception of concealing pregnancy) and have effects similar in magnitude to formal medical staff, and sometimes exceeding them (e.g., ANC registration). ASHAs also have strong positive associations with behaviors that occur early postpartum, even though many women did not receive a home visit in the first week after birth. Furthermore, ASHAs have relatively strong influences on behaviors that are directly incentivized, indirectly incentivized, and even those that are non-incentivized. We also investigated the possibility that ASHA contacts were moderated by differences between ASHAs and mothers in caste, religion, or wealth. While we did find some small moderating effects, the picture in Fig 7 implies that such differences are not responsible for the lack of ASHA reach to some households. That said, caste dynamics in rural Bihar have many effects on daily life and the analysis here should be interpreted cautiously, and not taken as an indicator that caste differences do not affect health system access.

Several lines of evidence suggest that ASHAs are highly salient in the minds of Bihari mothers across a broad spectrum of health-relevant behaviors. The regression for the overall effect of ASHAs on recommended health behavior (Fig 3) represents a highly general, emergent, positive association with uptake across 12 different behaviors. The mentions of the ASHA by

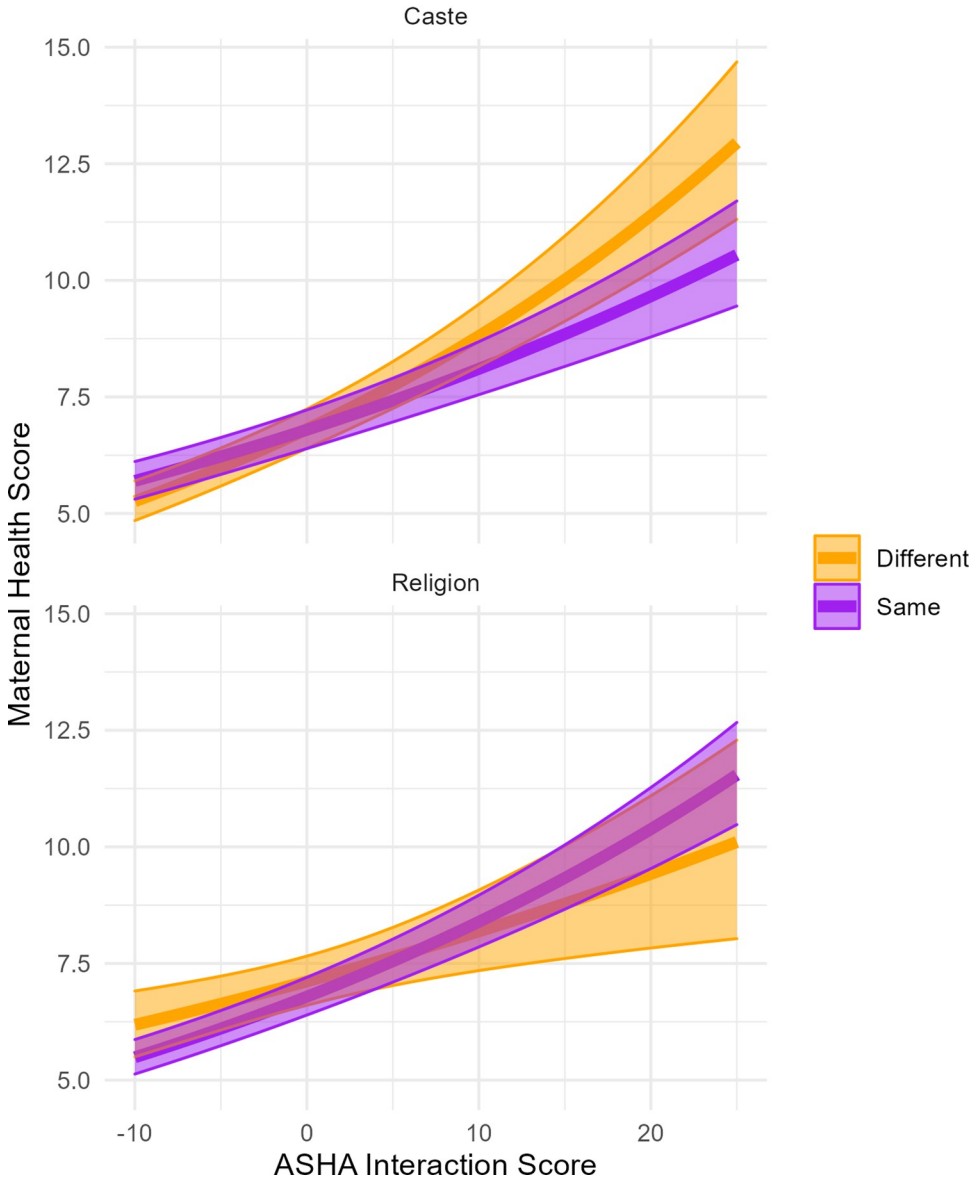

**Fig 7. Visualizing the moderating effects of caste and religion on the relationship between ASHA interaction score (mean-centered) and maternal health score.**

recent mothers in association with each of these behaviors is spontaneous; they are asked only about the general source of influence, not provided with a list of options. This general salience is similar to the regressions that capture the effect of ASHA interaction on maternal health score (Fig 2), but using a different measure based on a count of actual ASHA-named interactions and services provided. These effects are different from finding a specific relation between a type of visit and a single outcome, which are more commonly the focus in evaluations of ASHA performance. Together, they speak to a pervasive and highly general ASHA effect in the community.

## High ASHA efficacy is coupled with limitations in reach

The descriptions of ASHA–mother contacts make it clear that this general efficacy does not have a sufficiently broad reach, however. ASHA guidelines do not specify a target number of visits during pregnancy, but they do stipulate that visits *should* occur. A recent analysis of ASHA performance in Uttar Pradesh by Smittenaar et al. [11] suggested that ASHAs should visit an expectant mother "as soon as possible after learning she is pregnant and 4–6 times over the course of the pregnancy." This suggestion seems reasonable and if taken as a target, a median number of visits would require an increase of 1 to 3 visits per pregnancy in this sample.

## Distinguishing ASHAs from the ASHA program

To disentangle the factors that might lead to high efficacy, on the one hand, and insufficient reach, on the other, we think it is important to distinguish between the ASHA-as-person and the ASHA program. As individuals, ASHAs have high efficacy, but efficacy could be better leveraged or enhanced with support and training from the program. In other words, conditional upon visiting the mothers or otherwise reaching them with messaging, the ASHAs are highly effective. Perhaps improvements could be made at the program level that find ways to increase the workforce, manage the workload, or remove barriers from under-serviced households. This needs to be kept in mind when discussing a need for ASHAs to increase home visits or other issues that also effectively increase demands on her time, travel requirements, and the like.

As a program, there may be certain factors that limit or constrain the ASHA workforce. For instance, transportation is a commonly mentioned difficulty of the job. ASHA catchment areas are based on approximate population estimates with a target of about 1000 people per ASHA, but nonetheless may vary from one ASHA to another. This variation in catchment size could inadvertently lead to some households being more difficult to reach than others. Another job-related factor for the ASHA is monthly or seasonal variation in the number of pregnant women within a catchment. In our data for 400 ASHAs, the number of pregnant women in a catchment area at the time of the survey varied from 0 to 41 (most had from 5 to 11) and likely varies sporadically (by season or chance) during the year, which could cause sudden jumps and reductions in workload that are beyond the ASHA's control. Another factor that affects ASHA reach is a common practice whereby mothers move back to their natal households to give birth, so they can be near their parents and immediate families. This indirectly affects ASHAs because expectant mothers are more likely to be acquainted with the ASHA's near their husband's family home. ASHA programs also routinely add duties to what ASHAs are asked to do in their communities. Examples of this include conducting household surveys for some other government health initiative (outside of typical ASHA duties) or assisting with drug administrations or camps/workshops on special topics. The number of additional tasks that ASHAs are asked to assist with likely increases during the COVID-19 pandemic [29].

Some studies, like the one by Kumar et al. [28] referenced above, aggregate related behaviors into units to capture a more complete picture of a health service. For example, their "Full ANC" consists of taking 100 or more IFA tablets, getting at least one TT injection, and having four or more ANC checkups. A metric like this is highly useful for assessing service access, in that to improve health across a region requires that individuals gain access to several services. However, a measure like this could be misleading if used uniquely for evaluating ASHA efficacy because it would combine factors at the program level with those at the individual level. IFA tablets may be in short supply, which is out of the ASHA's ability to control. The mother

may decide to register late for ANC care, which the ASHA may or may not be able to influence and may require some skills in persuasion rather than just service extension, but just the same is not fully under her control and is different from a supply issue. TT injections may involve negative misconceptions associated with vaccines, or quality concerns for drugs distributed free of cost, both of which are tied to perceptions and misinformation that the ASHA may try to persuasively combat but also go well beyond what she can directly control, keeping in mind she has low pay and limited training. ANC checkups are visits out of the house and some mothers and their families may avoid these visits due to fear of evil eye, while some mothers may not have available family members to accompany them to the facility [15]. These are some of the various factors that can differentially impact the constituent parts of a composite score like Full ANC, and further indicate the need to distinguish programmatic factors from individual ones. Many researchers are seeing a need to shift the ASHA's role more toward a facilitator (or activist), which also requires various forms of improved programmatic support [7, 9, 11, 30, 31].

## Home visits are valuable but ASHAs may have other routes for messaging

Many women are receiving ASHA visits outside the home or are having information sent to them by ASHAs via family members. These other messaging routes may be effective and help buffer the effects of seemingly less than ideal frequencies of home visits. Indeed, finding multiple ways to meet with mothers is encouraged in the ASHA guidelines and one would expect that finding multiple messaging strategies via one's intuitive understanding of the community might be a skill that embedded CHWs develop over time. Raising this possibility is not intended to suggest that home visits are not valuable or critical, but rather to suggest that ASHAs may have ways to get their messages into the cultural system without a formal visit, which could be useful if there are occasionally barriers to making home visits. Additionally, evaluations of ASHA performance that target home visits should also consider other kinds of contacts, even if they occur outside of the home, especially since these are encouraged in ASHA guidelines.

## The measurements of ASHA efficacy sometimes conflict with the intention of the ASHA program

As mentioned above, several previous evaluations of ASHAs or the ASHA program have found mixed and negative results. Across the papers summarized in a recent thorough review by Scott et al. [14] are a variety of interesting and important studies that vary greatly from one another. This variation makes it difficult to form general impressions about the overall effect of the ASHA program. On the question of mixed results, however, we would like to note a few points from our study. One is that the effect of the ASHA on behavioral uptake is positive, given that efficacy is evaluated as raising the probability that mothers practice recommended health behaviors.

Perhaps in some classification schemes, the results reported here could be characterized as 'mixed' because we show that ASHAs could be "under-performing" if performance is measured by meeting a recommended target number of ANC home visits or by ensuring that ASHA services reach 100% of the mothers in an ASHA's catchment area. That is, we find ASHA contacts have a strong positive influence, but that the contacts may not be sufficiently widespread. If so, there could be ways to amplify these positive effects by finding ways to increase the frequency or reach of these visits, or in finding other ways for ASHAs to get messages to mothers. This begs the question of how many more ANC visits would it take to reach some kind of ceiling-effect or diminishing return with respect to the increased uptake of

recommended behaviors by mothers. It also begs the question of what is responsible for the association between ASHAs and the uptake of recommended behaviors. Direct visits during pregnancy are clearly a valuable messaging opportunity. The earlier the first home visit occurs, the more likely it is that women complete their IFA treatments, for example, which is partly why we consider not concealing the pregnancy from ASHAs as a biomedically-recommended behavior. It may also be the case that as the ASHA becomes more present and her messaging permeates the cultural milieu of a village, many women may associate the ASHA with the bio-medically-recommended behaviors even if they have not had many one-on-one visits from an ASHA. Lastly, if an ASHA's efficacy is going to be considered mixed or negative based on a number of visits of a given type, there needs to be clear evidence that more visits of that type would be optimal or that better alternatives to the type do not exist, which is a matter for further research. If a greater emphasis is placed on increasing ASHA home visits during pregnancy in the absence of further programmatic support, what allowances are there that some other service will decline in frequency as a result?

Like many CHW programs, the ASHA program is built on the notion that there is something cost-effective and outcome-promoting from attempting to recruit local workers who share customs and beliefs with their communities. Evaluations of either individual-level ASHA performance or of the ASHA program should keep this foundational rationale in mind. We suggest that evaluations should be more calibrated to what ASHAs actually do, given resources, training, and programmatic constraints. Evaluations should be designed with a rationale that complements that of the ASHA program.

## For further study

There is a broad association between ASHAs and generally health-positive advice. For that advice to reach critical thresholds, it has to spread through various channels, and it may be that an under-tapped aspect of CHWs is relying on their abilities to recognize and effectively utilize those channels as they grow into their roles. Training programs could be updated to facilitate that, ideally in collaboration with experienced ASHAs. Likewise, Figs 5 and 6 show that the ASHA is one among many sources of influence, many of which are based in the community or even the household. Finding ways to leverage alliances among these sources of influence may be a key to shifting the ASHA from service extender to cultural facilitator [32, 33].

## Conclusion

Many previous evaluations of ASHA efficacy have either not included actual measures of perinatal health behaviors or have looked at a fairly narrow range of them. Here we consider 12 perinatal health behaviors and 9 potential sources of influence. ASHAs have a positive influence on all 12 of these behaviors. In every case, they are among the influencers with the strongest positive effect, are sometimes the ones with the largest effect, and always have a statistically positive effect. We conclude that interactions between ASHAs and mothers positively impact mothers to engage in health-promoting behaviors. Our data are consistent with recommendations to increase ASHA contact with mothers, as well as improve ASHA training to more effectively educate mothers about health-promoting behaviors. We also encourage more studies that attempt to move beyond simple linear connections between an incentive or a particular kind of visit and a single outcome. Finally, we recommend further attention be paid to understanding how the efficacy of ASHAs can be further strengthened by matching mechanisms and sources of influence with particular focal behaviors [11]. ASHAs are effective catalysts of behavior change but increasing their efficacy further will require changes at the program level.

## Supporting information

**S1 Fig. Percent of mentions for each behavior associated with each potential source of influence.** Behaviors are listed roughly sequentially from top to bottom on the y-axis. Influencers are ordered by the overall frequency of mention from left to right on the y-axis such that more frequently mentioned influencers are to the left.
(TIF)

**S2 Fig. Results of moderation analysis.** Coefficients for each control variable, potential moderator (wealth, caste, and religion), ASHA interaction score, and interactions between moderators and ASHA interaction score.
(TIF)

**S1 Table. Counts for yes/no questions on the ASHA interaction score.**
(DOCX)

**S2 Table. Logistic regression for overall effect of each influencer on the uptake of the 11 focal behaviors.**
(DOCX)

**S3 Table. Results of selected logistic regression model that includes controls, influencers, behaviors, and influencer x behavior interactions.**
(DOCX)

**S4 Table. Results of moderation analysis that includes controls and an interaction between each moderator and ASHA interaction score.**
(DOCX)

**S5 Table. Results of moderation analysis that includes controls and an interaction between each moderator and ASHA interaction score.**
(DOCX)

## Acknowledgments

This manuscript and the broader project that it is associated with (Project RISE) are collaborative and were supported many talented individuals. We especially thank the investigators for PCI India, based in Patna, who collected data for the project and the many mothers who provided their time and information during the surveys.

## Author Contributions

**Conceptualization:** Oskar Burger, Maciej J. Dańko, Santosh Akhauri, Indrajit Chaudhuri, Emily Little, Sudipta Mondal, Neela Saldanha, Palash Singh, Tracy Johnson.

**Formal analysis:** Oskar Burger, Maciej J. Dańko.

**Funding acquisition:** Emily Little, Cristine H. Legare.

**Investigation:** Faiz Hashmi, Indrajit Chaudhuri.

**Methodology:** Oskar Burger, Faiz Hashmi, Santosh Akhauri, Emily Little, Tracy Johnson.

**Project administration:** Oskar Burger, Faiz Hashmi, Santosh Akhauri, Hannah G. Lunkenheimer.

**Writing – original draft:** Oskar Burger.

**Writing – review & editing:** Faiz Hashmi, Maciej J. Dańko, Santosh Akhauri, Emily Little, Hannah G. Lunkenheimer, Sudipta Mondal, Nachiket Mor, Neela Saldanha, Janine Schooley, Palash Singh, Tracy Johnson, Cristine H. Legare.

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
