## [Decision Letter · Decision Letter 0]

1 Mar 2022

PGPH-D-21-01173

Facilitating behavioral change: A comparative assessment of ASHA efficacy in rural Bihar

Dear Dr. Burger,

Thank you for submitting your manuscript to PLOS Global Public Health. After careful consideration, we feel that it has merit but does not fully meet PLOS Global Public Health’s publication criteria as it currently stands. Therefore, we invite you to submit a revised version of the manuscript that addresses the points raised during the review process.

Please note that the second reviewer has shared a marked PDF with comments left at various parts of the manuscript embedded within the PDF. 

We look forward to receiving your revised manuscript.

Kind regards,

Prashanth Nuggehalli Srinivas, MBBS, MPH, PhD

Academic Editor

Journal Requirements:

1. Please amend your detailed Financial Disclosure statement. This is published with the article, therefore should be completed in full sentences and contain the exact wording you wish to be published.

i). State the initials, alongside each funding source, of each author to receive each grant.

ii). State what role the funders took in the study. If the funders had no role in your study, please state: “The funders had no role in study design, data collection and analysis, decision to publish, or preparation of the manuscript.”

iii). If any authors received a salary from any of your funders, please state which authors and which funders.

2. Please ensure that the funders and grant numbers match between the Financial Disclosure field and the Funding Information tab in your submission form. Note that the funders must be provided in the same order in both places as well.

3. Please update your Competing Interests statement. If you have no competing interests to declare, please state: “The authors have declared that no competing interests exist.”

4. In the online submission form, you indicated that your data will be submitted to a repository upon acceptance.  We strongly recommend all authors deposit their data before acceptance, as the process can be lengthy and hold up publication timelines. Please note that, though access restrictions are acceptable now, your entire data will need to be made freely accessible if your manuscript is accepted for publication. This policy applies to all data except where public deposition would breach compliance with the protocol approved by your research ethics board. If you are unable to adhere to our open data policy, please kindly revise your statement to explain your reasoning and we will seek the editor's input on an exemption. Please be assured that, once you have provided your new statement, the assessment of your exemption will not hold up the peer review process.

5. Please provide separate figure files in .tif or .eps format only and remove any figures embedded in your manuscript file. Please also ensure that all files are under our size limit of 20MB.

6. We have noticed that you have uploaded supporting information but you have not included a list of legends.  Please add a full list of legends for all supporting information files (including figures, table and data files) after the references list.

Additional Editor Comments (if provided):

Reviewers' comments:

Reviewer's Responses to Questions

**Comments to the Author**

1. Does this manuscript meet PLOS Global Public Health’s publication criteria? Is the manuscript technically sound, and do the data support the conclusions? The manuscript must describe methodologically and ethically rigorous research with conclusions that are appropriately drawn based on the data presented.

Reviewer #1: Partly

Reviewer #2: Partly

2. Has the statistical analysis been performed appropriately and rigorously?

Reviewer #1: Yes

Reviewer #2: Yes

3. Have the authors made all data underlying the findings in their manuscript fully available (please refer to the Data Availability Statement at the start of the manuscript PDF file)?

Reviewer #1: Yes

Reviewer #2: Yes

4. Is the manuscript presented in an intelligible fashion and written in standard English?

Reviewer #1: No

Reviewer #2: Yes

5. Review Comments to the Author

Reviewer #1: Dear Authors,

Thank you for taking the time to do this elaborate study - on an important topic. I commend you on taking up a very elegant study design.

My concerns are primarily with the organization of the paper and the readability of it.

I encourage you to organize the material better-

a) The introduction is too elaborate- can you cut it down and emphasize why specifically you did this study?

b) With the elegant study design - I got lost in the language and the places where the methods and results merged. I would suggest a section called "Instrumentation" - that crisply describes the process and the specific survey tools developed.

- Please refrain from adding results to this.

c) The results section is confusing partly because of too many tables/figures. Could you simplify these? It would enhance the readability significantly.

d) Please do not bring in discussion elements into the results section.

e) The discussion section has a lot of richness. I wish it were more succinct. I felt like I was reading an elaborate results section rather than a discussion section.

Again, there is a lot of richness in this manuscript. This is a field I am very familiar with. However, it is a very difficult one to read. I would strongly recommend simplifying the writing to reach your readers.

Reviewer #2: Hi, Please find attached the manuscript with my comments as ‘sticky notes’ in the attached document.

My over all comment: publishable only after these comments are addressed. Also, need to simplify the way the graphs are explained in results. And not attribute causation.

6. PLOS authors have the option to publish the peer review history of their article (what does this mean?). If published, this will include your full peer review and any attached files.

**Do you want your identity to be public for this peer review?** For information about this choice, including consent withdrawal, please see our Privacy Policy.

Reviewer #1: No

Reviewer #2: **Yes: **Somesh Kumar

---

## [Editor Report · Decision Letter 1]

16 Jun 2022

Facilitating behavioral change: A comparative assessment of ASHA efficacy in rural Bihar

PGPH-D-21-01173R1

Dear Burger,

We are pleased to inform you that your manuscript 'Facilitating behavioral change: A comparative assessment of ASHA efficacy in rural Bihar' has been provisionally accepted for publication in PLOS Global Public Health.

Best regards,

Julia Robinson

Executive Editor